# Using a Paleo Ratio to Assess Adherence to Paleolithic Dietary Recommendations in a Randomized Controlled Trial of Individuals with Type 2 Diabetes

**DOI:** 10.3390/nu13030969

**Published:** 2021-03-17

**Authors:** Alexander Mårtensson, Andreas Stomby, Anna Tellström, Mats Ryberg, Maria Waling, Julia Otten

**Affiliations:** 1Department of Public Health and Clinical Medicine, Umeå University, 90187 Umeå, Sweden; martensson.alexander@hotmail.com (A.M.); andreas.stomby@umu.se (A.S.); annatellstrom30@gmail.com (A.T.); mats.ryberg@umu.se (M.R.); 2Region Jönköping County, 55592 Jönköping, Sweden; 3Department of Food, Nutrition and Culinary Science, Umeå University, 90187 Umeå, Sweden; maria.waling@umu.se

**Keywords:** Paleolithic diet, type 2 diabetes, dietary intervention, triglycerides, blood pressure, weight loss

## Abstract

This study is a secondary analysis of a randomized controlled trial using Paleolithic diet and exercise in individuals with type 2 diabetes. We hypothesized that increased adherence to the Paleolithic diet was associated with greater effects on blood pressure, blood lipids and HbA1c independent of weight loss. Participants were asked to follow a Paleolithic diet for 12 weeks and were randomized to supervised exercise or general exercise recommendations. Four-day food records were analyzed, and food items characterized as “Paleolithic” or “not Paleolithic”. Foods considered Paleolithic were lean meat, poultry, fish, seafood, fruits, nuts, berries, seeds, vegetables, and water to drink; “not Paleolithic” were legumes, cereals, sugar, salt, processed foods, and dairy products. A Paleo ratio was calculated by dividing the Paleolithic calorie intake by total calorie intake. A multiple regression model predicted the outcome at 12 weeks using the Paleo ratio, group affiliation, and outcome at baseline as predictors. The Paleo ratio increased from 28% at baseline to 94% after the intervention. A higher Paleo ratio was associated with lower fat mass, BMI, waist circumference, systolic blood pressure, and serum triglycerides at 12 weeks, but not with lower HbA1c levels. The Paleo ratio predicted triglyceride levels independent of weight loss (*p* = 0.046). Moreover, an increased monounsaturated/saturated fatty acids ratio and an increased polyunsaturated/saturated fatty acids ratio was associated with lower triglyceride levels independent of weight loss. (*p* = 0.017 and *p* = 0.019 respectively). We conclude that a higher degree of adherence to the Paleolithic diet recommendations improved fat quality and was associated with improved triglyceride levels independent of weight loss among individuals with type 2 diabetes.

## 1. Introduction

Type 2 diabetes is a common disease and a growing public health issue worldwide [1]. The disease increases the risk of cardiovascular morbidity and mortality, which can be reversed by risk factor control with lifestyle intervention [2]. A healthy diet is recommended and an important step in improving cardiovascular health [3]. However, scientific evidence is inconclusive, as different diets, e.g., the ketogenic and the Mediterranean diet, improve cardiovascular risk factors and glucose control [4,5]. Generally, a low intake of fat and a moderate intake of carbohydrates, preferably unrefined, is advised [3,6]. Evidence suggests beneficial effects of limiting added sugar intake, replacing saturated fats with unsaturated fats, and limiting calorie intake in the case of patients who are overweight. These recommendations are also given to individuals with type 2 diabetes [7,8].

A Paleolithic diet is based on what anthropological research has suggested was eaten during the Paleolithic age, but geographical location had a large effect on food availability [9,10].

Some reports suggest that the Paleolithic diet could be effective for the prevention and treatment of type 2 diabetes [10,11]. The diet consists of lean meat, poultry, fish, shellfish, vegetables, fruits, berries, eggs, nuts, and seeds. Legumes, dairy products, cereals, sugar, salt, and processed fats are excluded. The Paleolithic diet does not include recommendations for specific macronutrient proportions. Thus, the ratio of macronutrients in various Paleolithic diets varies substantially: 28–58% fat, 22–40% carbohydrates, and 9–35% protein [12]. 

In most intervention studies with the Paleolithic diet macronutrient composition is not fixed but diet composition is studied instead [10,13]. The Paleolithic diet has been shown to improve glucose tolerance, blood pressure, plasma insulin, and plasma triglyceride levels independent of decreased weight and waist circumference in patients with ischemic heart disease, as well as healthy individuals [14,15]. In obese postmenopausal women, the Paleolithic diet had a higher impact on weight and triglycerides than a diet based on the Nordic Nutrition Recommendations [16]. Short-term interventions ranging from 10 days to 12 weeks have also shown positive effects on glycaemic control and cardiovascular risk factors in both individuals with type 2 diabetes and in a healthy population [17,18,19]. 

A meta-analysis summarized the results of eight randomized controlled trials that compared the Paleolithic diet to various other recommended diet, e.g., the American diabetes association recommended diet, the Nordic nutrition recommendations and the Mediterranean like diet [20]. The meta-analysis showed that a Paleolithic diet promotes weight loss, lowers BMI and waist circumference, and improves fat mass, systolic blood pressure (SBP), and diastolic blood pressure (DBP), as well as triglycerides, LDL, and HDL cholesterol when compared to other healthy diets [20]. Adherence to the diet can modify the effects on these outcomes [21]. However, most dietary interventions rarely report adherence to the diet and simply state that reporting bias could have affected the results. 

In a previous study, subjects with type 2 diabetes were asked to follow a Paleolithic diet and were randomized to either supervised exercise or not for 12 weeks. Participants in this study decreased their weight, fat mass, blood pressure, triglycerides, HbA1c, and fasting glucose as previously reported [22]. However, whether the degree of adherence to the Paleolithic diet recommendations predicted outcomes at the end of the intervention is unclear. Therefore, the aim of the present study was to elucidate whether the degree of adherence to the Paleolithic diet modified the intervention outcome with respect to BMI, fat mass, blood pressure, triglycerides, HbA1c, and fasting glucose.

## 2. Materials and Methods

### 2.1. Study Design

In this randomized controlled trial, 32 individuals with type 2 diabetes were asked to follow a Paleolithic diet for 12 weeks. Participants were randomized to either a supervised exercise regimen three times per week which constituted the exercise group. The other group were given general recommendations to perform 30 min of unsupervised exercise daily, as recommended for all individuals with type 2 diabetes (diet only group). Measurements from baseline and 12 weeks were analyzed. The secondary analysis reported in this paper is an observational study investigating the association between different measurement in a regression analysis. Moreover, a non-randomized observational group was examined twice as a reference.

### 2.2. Study Population

Study participants were recruited in 2012 from the surrounding areas of Umeå, Sweden. Inclusion criteria were type 2 diabetes (duration < 10 years) with either dietary treatment and/or metformin, BMI 25–40 kg/m^2^, <5% change in weight during the last 6 months, and age 30–70 years for males and postmenopausal up to 70 years of age for women. Exclusion criteria were blood pressure >160/100 mmHg, treatment with beta-blockers, macroalbuminuria, heart disease, smoking, or the use of any antidiabetic drugs other than metformin. Participants’ baseline characteristics are reported in Table 1.

Inclusion criteria for the secondary analysis of this present study was that the participant registered a 4-day self-reported, weighed, food record at both baseline and the endpoint of the study. One participant had no metabolic outcomes recorded but was included in the analysis of anthropometric measurements. The study was approved by the Regional Ethical Review Board, Umeå, Sweden (Dnr2011-294-31M), registered at clinicaltrials.gov (NCT01513798). All participants gave written informed consent. 

### 2.3. Dietary Intervention

All the participants were instructed to eat a Paleolithic diet as shown in Table 2. Food items considered Paleolithic were lean meats, poultry, fish, seafood, fruits, nuts, berries, seeds, vegetables, and mainly water to drink. Food items that were excluded from the Paleolithic diet included legumes, cereals, sugar, salt, processed foods, and dairy products. Participants were given information about the Paleolithic diet, different recipes, practiced in cooking sessions and were encouraged to ask questions about the diet. The intervention is described in detail in a previous paper [22]. 

### 2.4. Food Records

Dietary intake was assessed using four-day self-reported weighed food records in which all foods, beverages, and leftovers were weighed and estimated at baseline and at 12 weeks. In certain cases—e.g., restaurant visits—the Swedish Food Agency’s portion guide was used to estimate the amounts of food eaten [23]. If a complete meal—e.g., a meat stew—was added to the record, the participants were asked by the project dietitian about the specific ingredients and proportions of the meal. Additional information, such as cooking methods and other comments, were added by the participants. Any questions regarding food records were addressed during meetings, via email, or by phone. The nutritional calculation program Dietist XP v.3.2 was used to calculate energy and nutrient intake from the food records of all participants using the database from the Swedish Food Agency (v.2011-02-14), which contains approximately 2000 different food items.

### 2.5. Calculation of the Paleo Ratio and Other Ratios

Foods included in the 4-day self-reported weighed food records were grouped into Paleolithic and non-Paleolithic foods as summarized in Table 2. In cases of uncertainty regarding whether a food item was Paleolithic, the food item was discussed with a qualified dietician and within the research group.

A Paleo ratio was calculated by dividing the energy content (kcal) of the Paleolithic food items by the total energy intake (kcal). The Paleo ratio was used to assess the proportion of Paleolithic food items in the diet at baseline and after 12 weeks. Only at 12 weeks participants were instructed to follow the Paleolithic diet. For example, if a participant ate 10,000 kcal for 4 days, with Paleolithic foods making up 6000 kcal, the Paleo ratio was calculated as 6000/10,000 kcal = 0.6 (e.g., 60% of the diet was Paleolithic). Using a ratio and analyzing a dietary pattern instead of analyzing specific foods or nutrients is considered a good approach when studying dietary effects on diseases [24].

A monounsaturated/saturated fatty acids ratio (MUFA/SFA ratio) was calculated by dividing MUFA (gram) by SFA (gram) based on the participants food records at baseline and at 12 weeks. The polyunsaturated/saturated fatty acids ratio (PUFA/SFA ratio) was based on the ratio of PUFA (gram) to SFA (gram) reported at baseline and at 12 weeks. A protein/carbohydrate ratio was calculated by dividing protein intake (gram) by carbohydrate intake (gram) based on the participants food records at 12 weeks. 

### 2.6. Body Composition, Anthropometric, and Biochemical Measurements

Fat mass was measured by dual energy X-ray absorptiometry (Lunar Prodigy X-ray Tube Housing Assembly, Brand BX-1 L, Model 8743; GE Medical Systems, Madison, WI, USA). BMI was calculated as weight (in kg) divided by height (in m) squared. Body weight was measured in light clothing on a calibrated digital scale. Height was measured using a calibrated height-measuring gauge. Waist circumference was measured using a measuring tape halfway from the lowest rib to the iliac crest as the participant gently exhaled. Blood pressure was measured in the sitting position after 5 min of rest using an automatic meter (Boso Medicus, Bosch, Germany). Serum triglycerides, serum cholesterol, and HbA1c were analyzed at the Department of Clinical Chemistry at Umeå University Hospital using fasting venous blood samples. Fasting glucose was analyzed in a capillary blood sample using HemoCue (HemoCue 201, RT; Radiometer Medical aps, Brønshøj, Denmark). 

### 2.7. Observational Group

We also report results from a small (*n* = 8) observational group that was not randomized but measured before and after 12 weeks. With this group we wanted to investigate if individuals change their dietary habits and/or decrease body weight, blood pressure or blood lipids when examined two times with 12 weeks in between. Participants of the observational group were recruited among those who were excluded from the intervention study because of different reasons. The observational group could not be included in the regression analysis because the Paleo ratio at the 12 weeks was not normaly distributed when both intervention groups and the observational group were included. The results of the 12-weeks-Paleo ratio of the observational group had no overlap with those of the intervention groups and data could not be transformed to achieve acceptable normal distribution.

### 2.8. Statistical Analysis

Data are reported as median and interquartile range if not otherwise stated. The change over time within in the same group was analyzed with the Wilcoxon signed ranks test. Multiple linear regression was used to test the association between the Paleo ratio at 12 weeks and the intervention effect on the outcome variable (e.g., fat mass, SBP, triglycerides) at 12 weeks. Because of the small sample size (*n* = 27), we limited the maximum number of independent variables in the regression model to three. In model 1, the regression analysis was adjusted for the relevant outcome variable at baseline. For example, we used the Paleo ratio at 12 weeks and adjusted for triglyceride levels at baseline to predict triglyceride levels at 12 weeks in regression model 1. For Model 1b, the regression analysis was adjusted for the relevant outcome variable at baseline as well as the Paleo ratio at baseline. In regression model 2, we investigated the effect of the paleo ratio at 12 weeks and adjusted for the outcome variable at baseline and group affiliation (Paleolithic diet group or Paleolithic diet and exercise group). For example, we used the Paleo ratio at 12 weeks adjusting for triglyceride levels at baseline and group affiliation to predict triglyceride levels at 12 weeks. In regression model 3, we investigated the effect of the Paleo ratio at 12 weeks adjusting for the outcome variable at baseline and change in weight. For example, we used the Paleo ratio at 12 weeks adjusting for triglyceride levels at baseline and change in weight to predict triglyceride levels at 12 weeks. Moreover, triglyceride levels and SBP were analyzed using the same multiple regression as in model 2, but substituting Paleo ratio for change in weight. In regression model 4, we investigated the effect of the MUFA/SFA ratio at 12 weeks adjusted for the outcome variable at baseline and group affiliation (Paleolithic diet group or Paleolithic diet and exercise group). For example, we used the MUFA/SFA ratio at 12 weeks adjusting for triglyceride levels at baseline and group affiliation to predict triglyceride levels at 12 weeks. In regression model 5, we investigated the effect of the PUFA/SFA ratio at 12 weeks adjusted for the outcome variable at baseline and group affiliation (Paleolithic diet group or Paleolithic diet and exercise group), e.g., using triglyceride levels at baseline, group affiliation and PUFA/SFA ratio at 12 weeks to predict triglyceride levels at 12 weeks.

The results of the regression models are reported with unstandardized B and its 95% confidence intervals. We have chosen unstandardized B instead of standardized B because we do not compare different outcome variables in one analysis. With unstandardized B the clinical effect of the Paleo ratio can be deduced directly from the analysis. A two-tailed *p* < 0.05 was considered significant. Data can be obtained from the corresponding author on reasonable request. Data cannot be made available on a public source based on privacy regulations. Statistical analyses were performed using IBM SPSS Statistics for Windows, Version 25 (IBM Corp., Armonk, NY, USA).

## 3. Results

As reported previously [22], both intervention groups had similar effects on body composition, anthropometric, and biochemical measurements. The median weight loss in both groups was 7.1 kg without difference between groups. A total of 27 of the participants completed 4-day, self-reported, weighed food records at baseline and 12 weeks and were eligible for this study. At baseline, the participants had a median Paleo ratio of 0.28 (IQR 0.22–0.38), indicating that 28% of the participants’ reported energy intake at baseline was from food items considered Paleolithic. After 12 weeks of intervention, the median Paleo ratio was 0.94 (IQR 0.85–1.0). The Paleo ratio for each participant at baseline and after 12 weeks of intervention is reported in Figure 1. The participants reported a calorie intake of 1770 kcal/day at baseline (IQR 1414–2125) and of 1373 kcal/day at 12 weeks (IQR 933–1812). The calorie intake of each participant separately is reported in Figure 2. 

A higher Paleo ratio at 12 weeks was associated with lower fat mass, BMI, waist circumference, SBP, and triglyceride levels after 12 weeks of intervention in regression model 2 (all *p* < 0.05, Table 3). The Paleo ratio after 12 weeks explained 89% of the lower fat mass and BMI after the intervention according to the regression model (adjusted R^2^ = 0.89 for model 1 and model 2). 

Approximately 60% of the variation in SBP after 12 weeks was explained by the Paleo ratio (adjusted R^2^ = 0.62 for model 1 and 0.61 for model 2). However, DBP was not associated with the Paleo ratio after 12 weeks. Moreover, approximately 40% of the triglyceride levels at 12 weeks was explained by the Paleo ratio (adjusted R^2^ = 0.41 for model 1 and 0.39 for model 2). Other blood lipid levels were not associated with the Paleo ratio after 12 weeks (Table 4).

Fasting insulin, fasting glucose, HbA1c, and lean mass at 12 weeks were not associated with the Paleo ratio at 12 weeks according to our regression model (Table 4).

In the regression models described above, group affiliation was never significantly associated with the outcome. We conclude that group affiliation does not seem affected the results described above.

Weight loss per se was associated with reduced SBP (B = 1.39, 95% CI 0.55–2.23, *p* = 0.002), but not triglyceride levels (B = 0.06, 95% CI 0.01–0.12, *p* = 0.092). When weight loss was added to model 2, the association between the Paleo ratio and the decrease in SBP during the intervention became non-significant (*p* = 0.119). The triglyceride levels remained significantly associated with the Paleo ratio when weight loss was added to model 3 (*p* = 0.046, Table 4).

As reported previously, participants of both groups decreased their intake of saturated fatty acids (SFA) and tended to increase their intake of monounsaturated fatty acids (MUFA) and polyunsaturated fatty acids (PUFA) [22]. The MUFA/SFA ratio increased in the two intervention groups between baseline and 12 weeks (*p* < 0.0001, Figure 3). Moreover, the PUFA/SFA ratio increased during the 12 weeks of intervention (*p* < 0.0001, Figure 4). We therefore investigated with a regression model if the MUFA/SFA ratio and PUFA/SFA ratio respectively had an impact on anthropometry, blood pressure, blood lipids and glucose homeostasis at study end (Table 3). A higher MUFA/SFA ratio was associated with lower triglycerides. When weight loss was added to this regression model, the association between the MUFA/SFA ratio was still significant (B = −0.42, 95% CI−0.75; −0.83, *p* = 0.017). The association between the MUFA/SFA ratio and triglycerides is therefore considered independent of weight loss. Also, a higher PUFA/SFA ratio was associated with lower triglycerides (Table 3). When weight loss was added to the regression model, the association between the PUFA/SFA ratio and triglycerides was still significant (B = −1.0, 95% CI−1.9; −0.2, *p* = 0.019). The association between the PUFA/SFA ratio and triglycerides is therefore considered independent of weight loss.

We have reported previously that study participants decreased their carbohydrate intake and tended to increase their protein intake in this study [22]. When investigating the association between the protein/carbohydrate ratio and our study outcomes we could not find any significant association (data not shown). 

We investigated if the Paleo ratio at baseline influences the results by adjusting model 1 also for the Paleo ratio at baseline (model 1b). After this additional adjustment, the association between the Paleo ratio at 12 weeks and SBP (B = −23.10, 95% CI−45.54; −0.65, *p* = 0.044) and triglycerides (B = −1.83, 95% CI−3.35, −0.32, *p* = 0.020), respectively, was unchanged. 

The observational group reported a median Paleo ratio of 0.27 (IQR 0.13–0.41) at baseline and a median Paleo ratio of 0.23 (IQR 0.11−0.28) after 12 weeks (*p* = 0.093 for the change over time). At baseline, they had a median BMI of 31.0 (IQR 30.1–34.1) and after 12 weeks a median BMI of 31.2 (IQR 29.9–33.9, *p* = 0.263 for the change over time). SBP was at baseline 136 mmHg (IQR 131−142) and after 12 weeks 149 mmHg (IQR 131–155, *p*= 0.028 for the change over time). Plasma-triglycerides at baseline were 2.3 mmol/L (IQR 1.2–2.5) and after 12 weeks 1.6 mmol/L (IQR 1.4–1.7, *p* = 0.362 for the change over time).

## 4. Discussion

This study suggests that a high degree of adherence to Paleolithic dietary recommendations is associated with a decrease in fat mass, body weight, waist circumference, systolic blood pressure, and triglyceride levels among patients with type 2 diabetes. Furthermore, the lower systolic blood pressure seems to be driven by the weight loss, whereas the triglyceride levels seem to be more strongly associated with dietary adherence especially improved fat quality. The unrandomized observational group did not increase their intake of Paleolithic food and did not lower body weight, blood pressure or blood lipids.

The median self-reported dietary content of Paleolithic food items before the intervention was 28%. This was expected because the traditional Nordic diet contains grain-based food items, as well as dairy products, which are excluded from the Paleolithic diet [25]. The Nordic diet is comparable to the diet of our observational group which had a Paleo ratio of 27% at baseline and of 23% after 12 weeks. The Paleo ratio in our intervention groups increased to 94% at 12 weeks, which indicates a high degree of adherence to the Paleolithic diet recommendations. To what degree this change is due to reporting bias remains to be determined and is a subject of debate [26]. Previous studies, including the primary analysis of this randomized controlled trial, have found beneficial effects of a Paleolithic diet on body composition, anthropometric, and biochemical measurements among overweight and obese individuals with and without type 2 diabetes [20,22]. This study supports and extends these findings by suggesting that the degree of adherence to a Paleolithic diet positively correlates with these outcomes. The effect on weight loss is important because a healthy weight is key to avoid cardiovascular complications among obese patients with type 2 diabetes [27]. 

In the present study we found that a high intake of Paleolithic food items for 12 weeks was associated with lower triglyceride levels. This finding is in accordance with other studies [27]. Notably, this association was independent of weight loss, which may indicate that the dietary composition causes these changes. The important dietary factor we found in regards to plasma triglycerides was the increased intake of MUFA and PUFA in relation to a decreased intake of saturated fat. This may be caused, at least in part, by the increased intake of fish, which doubled after 12 weeks. Fish contains a high amount of omega-3 fatty acids, which have been shown to lower triglyceride levels [28]. A decreased intake of saturated fat and a higher consumption of MUFA and PUFA has been reported from a hunter-gatherer diet in Eskimos [29,30]. In this population, plasma triglyceride levels were more than 50% lower compared to Danish people eating a normal Western diet. The authors believe that zero intake of cereals and sugars and a high intake of meat especially from whale and seal may be the reason for the low plasma triglyceride levels [29,30]. In our study, we hypothesized that the lower carbohydrate intake in part explains the lower triglyceride levels [31]. However, we had to reject this hypothesis because the protein/carbohydrate ratio was not associated with plasma triglyceride levels at study end. Several other studies have found lower triglyceride levels after Paleolithic diet interventions, with a remaining effect after 2 years in overweight postmenopausal women without type 2 diabetes [15,16,17,27,32,33]. Similar results of lower triglyceride levels have also been found in other studies, including in Australian aborigines eating a hunter-gatherer diet [14,34]. Notably, increased triglyceride levels are a hallmark of insulin resistance and common among patients with type 2 diabetes [35]. This may explain why a high degree of adherence to the Paleolithic diet was associated with lower triglyceride levels but not altered LDL or HDL cholesterol levels. Moreover, several participants were treated with statins, which may attenuate the dietary effects on cholesterol levels [36]. Importantly, the Paleo ratio was not associated with decreased fasting insulin levels and, therefore, the association with lower triglyceride levels was not likely caused by effects on insulin resistance. 

Previous studies on the effects of a Paleolithic diet on blood pressure have been contradictory. In the primary analysis of this study published previously, we found a significant decrease in both systolic and diastolic blood pressure regardless of exercise [22]. On the other hand, Ryberg et al. demonstrated that diastolic but not systolic blood pressure, was significantly lower after 5 weeks of Paleolithic diet among women without type 2 diabetes [32]. Some other studies have also suggested that a Paleolithic diet reduces blood pressure [17,20], whereas others have not [14]. Our analysis suggests that weight loss, rather than adherence to the Paleolithic diet per se, is causing this improvement. The conclusion that weight loss has larger effects on blood pressure than dietary composition is supported by some previous studies [34,37,38], but not all [39] and the decreased calorie intake might have affected our results (Figure 2). Notably, 19 participants in this study were treated with anti-hypertensive drugs, which could have modified this outcome. The effects of a Paleolithic diet on hypertension must be elucidated further. In this study, HbA1c decreased significantly in both groups [22]. However, this effect was not related to a higher degree of adherence to the Paleolithic diet. Some studies have shown that HbA1c decreases more with a Paleolithic diet than with a specific diabetes diet, and that this is not explained by changes in body weight [17,18]. In our study, weight loss was not associated with improvements in HbA1c (data not shown). However, a meta-analysis showed that weight loss is a major factor in improving glucose tolerance and in the prevention of developing type 2 diabetes [40,41]. Thus, previous studies suggest that weight loss is most likely the main contributor to improved glucose levels, but this could not be confirmed in this study.

Indexes for describing dietary adherence, such as a Mediterranean diet index and a “Healthy diet index”, are primarily used in epidemiological studies and have been associated with lower morbidity and mortality [42,43,44]. In this study, the Paleo ratio was based on self-reported food records, whereas epidemiological studies usually use questionnaires. Using questionnaires to generate an index has limitations, such as a limited selection of food items and portion sizes [45]. Nevertheless, different indexes and the use of questionnaires seem to perform well in explaining diet adherence [46]. Weighed food records also have limitations but are considered the gold standard for assessing dietary intake in smaller populations. It is beneficial to use weighed food records because it does not rely on memory or the ability to estimate portion sizes. Although the method for weighed food records varies between studies, they perform evenly in predicting participants’ dietary intake [47]. 

All dietary assessment methods have measurement biases, such as underreporting, and there is always a risk that participants eat in a more favorable way during the days the diet is assessed. In the present study, the participants were instructed to report their habitual intake at baseline and 12 weeks according to the instructions, but this cannot be guaranteed and must be considered in the interpretation of the results. It is also possible that the 4 days of food recording did not capture the full variation in the participants’ diets [48]. However, adding more than 4 days to a food record increases the risk of other confounders, such as underreporting due to the added time and effort of keeping a food record. 

Another limitation is that the Paleolithic diet of this study was not exactly what inhabitants of this region ate during the Paleolithic age. In stone age, diet was largely affected by geographical location and seasonable availability of food items. However, in this study the Paleolithic diet included food items not native to Europe, e.g., cocoa, and most of the food was produced by agriculture and animal domestication. These adaptations were necessary for practical reasons and the Paleolithic diet described in this paper is therefore a modern adaptation of the genuine Paleolithic diet. Moreover, some food items, such as ham, contain salt and are typically not included in a Paleolithic diet but were allowed in this study to facilitate dietary adherence. As limiting salt is an important component of the Paleolithic diet, this may have reduced the dietary effects. Some food items were considered Paleolithic up to a certain amount, e.g., nuts 60 g/day. Thus, these food items were considered non-Paleolithic if the amount exceeded the threshold for maximum intake (Table 2). Consequently, eating too much of these food items may have decreased, rather than increased, the Paleo ratio.

The study included more men than women, in part because of more strict exclusion criteria for women. This unbalance might have affected our results. The observational group was not randomized and could not be included in the regression analysis because of statistical reasons. 

## 5. Conclusions

A higher degree of adherence to a Paleolithic diet was associated with beneficial effects on fat mass, body weight, waist circumference, systolic blood pressure and triglycerides. Although the reduction of systolic blood pressure seems to be mainly caused by weight loss, improvements in triglyceride levels were shown regardless of weight loss and may be caused by improved fat quality.

This suggests that focusing on strategies to improve dietary adherence is important for successful dietary interventions among individuals with type 2 diabetes. Future research focusing on methods for improving dietary adherence would be of great importance.

## Figures and Tables

**Figure 1 nutrients-13-00969-f001:**
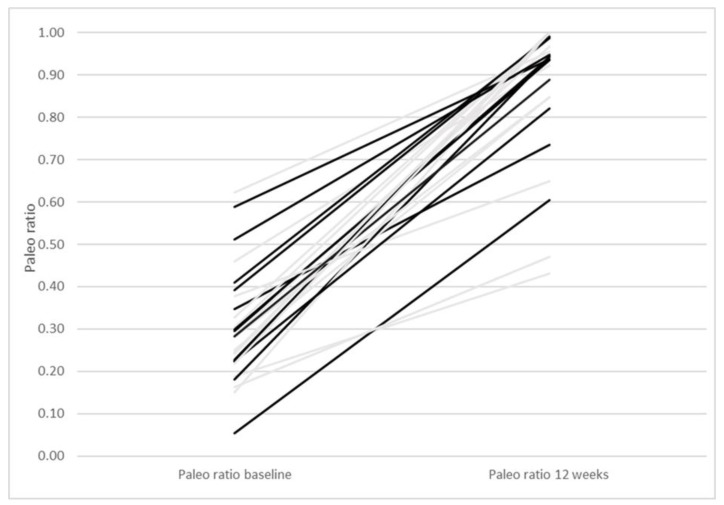
Paleo ratio with one line per participant starting with the Paleo ratio at baseline and ending at Paleo ratio at 12 weeks. Grey lines represent participants of the Paleolithic diet only group and black lines participants of the Paleolithic diet and exercise group.

**Figure 2 nutrients-13-00969-f002:**
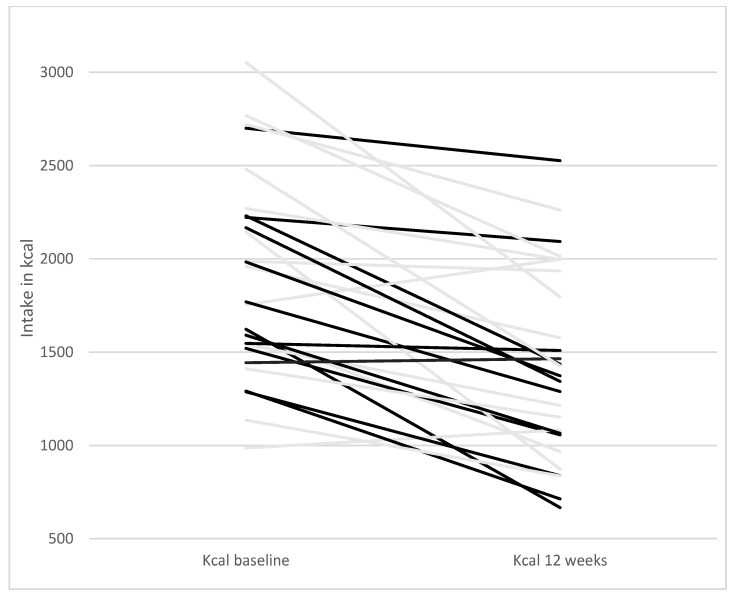
Daily calorie intake at baseline and 12 weeks. One line per participant starting with kcal intake per day at baseline and ending with kcal intake per day at 12 weeks. Grey lines represent participants of the Paleolithic diet only group and black lines participants of the Paleolithic diet and exercise group.

**Figure 3 nutrients-13-00969-f003:**
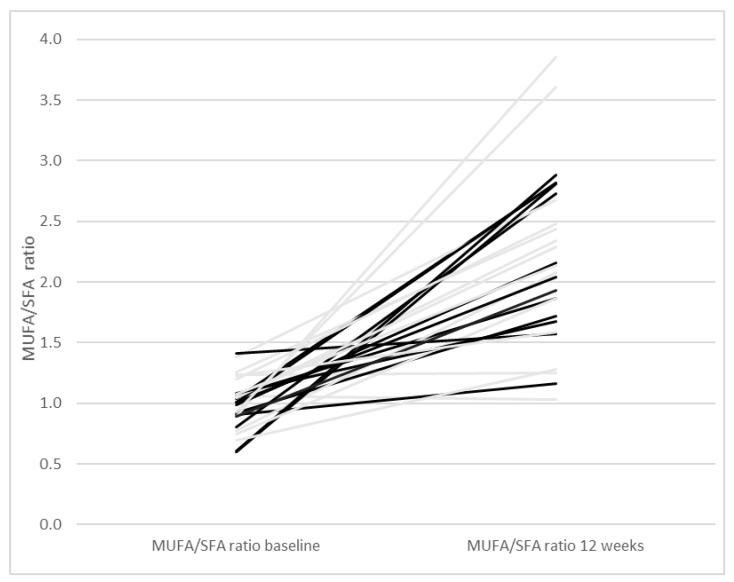
Monounsaturated fatty acids/saturated fatty acids ratio (MUFA/SFA ratio) with one line per participant starting with the MUFA/SFA ratio at baseline and ending at the MUFA/SFA ratio at 12 weeks. Grey lines represent participants of the Paleolithic diet only group and black lines participants of the Paleolithic diet and exercise group.

**Figure 4 nutrients-13-00969-f004:**
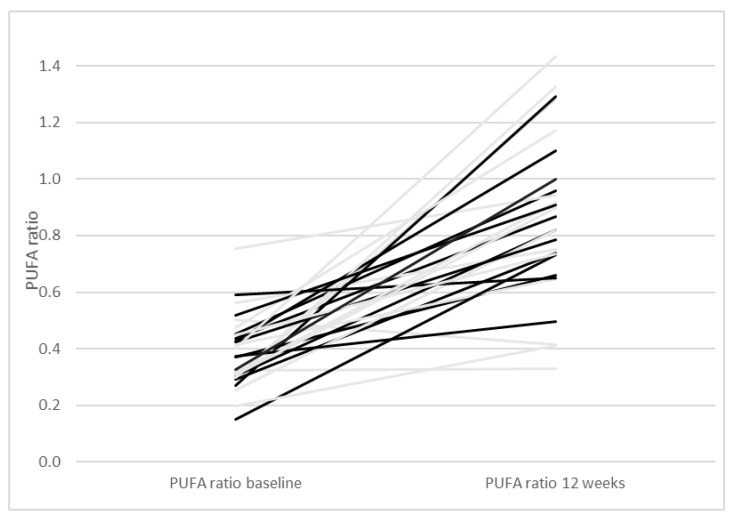
Polyunsaturated fatty acids/saturated fatty acids ratio (PUFA/SFA ratio) with one line per participant starting with the PUFA/SFA ratio at baseline and ending at the PUFA/SFA ratio at 12 weeks. Grey lines represent participants of the Paleolithic diet only group and black lines participants of the Paleolithic diet and exercise group.

**Table 1 nutrients-13-00969-t001:** Characteristics of participants.

	Paleolithic Diet (Only Diet) (*n* = 14)	Paleolithic Diet (Diet + Exercise) (*n* = 13)	Paleolithic Diet (All Participants) (*n* = 27)
Age (years)	61 (52–64)	61 (58–67)	61 (57–65)
Women/Men (*n*)	5/9	4/9	9/18
Diabetes duration (years)	3 (1–6)	6 (1–8)	3 (1–8)
BMI (kg/m^2^)	31 (29–34)	31 (29–35)	31 (29–33)
Hba1c (mmol/mol, *n* = 26)	54 (48–57)	57 (50–66)	55 (48–58)
Diabetes treatment (*n*)			
Metformin	10	9	19
No medication	4	4	8
Other treatments (*n*)			
ACEI or ARB	9	9	18
Diuretics	6	4	10
Calcium channel blocker	4	4	8
Statin	6	7	13
Antiplatelet drug	2	3	5
Other	7	2	9

Data reported as median (IQR). Abbreviations: ACEI; ACE inhibitor; ARB; angiotensin receptor blocker.

**Table 2 nutrients-13-00969-t002:** Food items considered Paleolithic or not with eventual restrictions.

Paleolithic Food Groups	Non-Paleolithic Food Groups
Berries	Beer
Cacao	Bouillon and salt
Fish/shellfish	Butter
Fresh juice	Bread
Fruits	Cheese
Honey	Fish/shellfish-based products (Ex; breaded fish)
Meats with added salt	Grains
Meats without added salt	Ice cream
Mushroom protein	Legumes
Mushrooms	Meat based products
Poultry	Milk/cream
Root vegetables (excluding potatoes)	Organ meats
Sauces (such as homemade mayonnaise)	Pasta
Vegetable products (including only items from Paleolithic food groups)	Preserves
Vegetables	Products containing sugar
Vinegar	Sauces (such as bearnaise)
	Sausages
	Snacks
	Soup
	Spirits
	Sweet drinks
	Vegetable dishes(e.g., vegetable dishes containing non-paleolithic food items)
	Vegetable products (e.g., salads with dairy)
	Yoghurt
**Paleolithic Food Included with Restrictions**	**Food Excluded**
Wine, one glass/week (150 g)	Wine over restricted amount
Rapeseed or olive oil, maximum 15 g/day	Rapeseed/olive oil over restricted amount
Potatoes, 1 medium sized/day	Potato-based products and potatoes exceeding restricted amount
Nuts, 60 g/day	Nuts above restricted amount
Eggs, 1–2/day, max 5/week	Egg based products and eggs exceeding restricted amount
Dried fruit, 130 g/day	Dried fruit exceeding restricted amount
Coffee/tea, 300 g/day	Instant coffee, or coffee exceeding restricted amount.

**Table 3 nutrients-13-00969-t003:** Association between the MUFA/SFA ratio and PUFA/SFA ratio and outcome variables after 12 weeks of intervention.

	MUFA/SFA Ratio		PUFA/SFA Ratio	
	B (95% CI)	*p*-Value	B (95% CI)	*p*-Value
**Anthropometry**				
Fat mass, kg	−1.2 (−2.5; 0.1)	0.075	−1.7 (−5.0; 1.6)	0.287
BMI, kg/m^2^	−0.6 (−1.1; 0.1)	0.080	−0.9 (−2.5; 0.7)	0.237
Waist circumference, cm	−1.9 (−4.2; 0.4)	0.108	−3.9 (−9.8; 1.9)	0.178
Lean mass, kg	−1.5 (−3.4; 0.5)	0.129	−2.7 (−7.5; 2.1)	0.260
**Blood pressure**				
SBP, mmHg	−4 (−10; 1)	0.099	−9 (−22; 4)	0.156
DBP, mmHg	0 (−3; 3)	0.931	−5 (−13; 3)	0.191
**Blood lipids**				
Triglycerides, mmol/L	−0.5 (−0.8; −0.2)	0.006	−1.1 (−1.9; −0.3)	0.011
HDL, mmol/L	0.0 (0.0; 0.1)	0.198	0.0 (−0.1; 0.2)	0.767
LDL, mmol/L	−0.1 (−0.4; 0.2)	0.630	0.1 (−0.6; 0.9)	0.717
**Glucose homeostasis**				
HbA1c, mmol/mol	−11 −7; 5)	0.788	−22 (−17; 13)	0.824
Fasting P-insulin, mlU/L	−1.8 (−4.4; 0.7)	0.154	−5.6 (−11.8; 0.5)	0.072
Fasting glucose, mmol/L	−0.2 (−1.2; 0.8)	0.691	−0.7 (−3.2; 1.9)	0.603

The table shows a linear regression analysis with unstandardized B and its 95% confidence intervals adjusted for the outcome measurement at baseline and group affiliation (Paleolithic diet group or Paleolithic diet and exercise group). Abbreviations: MUFA, monounsaturated fatty acids, SFA, saturated fatty acids, PUFA, polyunsaturated fatty acids. BMI, body mass index. SBP, systolic blood pressure. DBP, diastolic blood pressure. HDL, High-density lipoprotein. LDL, Low-density lipoprotein.

**Table 4 nutrients-13-00969-t004:** Association between the Paleo ratio and outcome variables after 12 weeks of intervention.

	Paleo Ratio—Model 1		Paleo Ratio—Model 2		Paleo Ratio—Model 3	
	B (95% CI)	*p*-Value	B (95% CI)	*p*-Value	B (95% CI)	*p*-Value
**Anthropometry**						
Fat mass, kg	−5.9 (−11.4; −0.5)	0.034	−5.5 (−11.0; −0.04)	0.048		
BMI, kg/m^2^	−2.6 (−5.2; 0.04)	0.053	−2.7 (−5.4; −0.02)	0.049		
Waist circumference, cm	−9.2 (−18.9; 0.5)	0.061	−9.8 (−19.6; −0.02)	0.0496		
Lean mass, kg	−0.8 (−6.5; 5.0)	0.786	−1.5 (−6.7; 3.7)	0.561		
**Blood pressure**						
SBP, mmHg	−23 (−44; −3)	0.029	−24 (−46; −3)	0.028	−15 (−35; 4)	0.119
DBP, mmHg	−1 (−14; 12)	0.865	0 (−14; 14)	0.958		
**Blood lipids**						
Triglycerides, mmol/L	−1.8 (−3.2; −0.4)	0.016	−1.7 (−3.2; −0.03)	0.022	−1.6 (−3.1; −0.03)	0.046
HDL, mmol/L	0.0 (−0.2; 0.3)	0.737	0.0 (−0.2; 0.3)	0.754		
LDL, mmol/L	−0.9 (−2.2; 0.4)	0.170	−0.9 (−2.2; 0.4)	0.183		
**Glucose homeostasis**						
HbA1c, mmol/mol	−6 (−30; 19)	0.645	−5 (−31; 20)	0.671		
Fasting P-insulin, mlU/L	−3.4 (−15.5; 8.8)	0.569	−3.2 (−15.5; 9.1)	0.594		
Fasting glucose, mmol/L	0.4 (−3.8; 4.7)	0.833	0.5 (−3.9; 4.9)	0.814		

The table shows a linear regression analysis with unstandardized B and its 95% confidence intervals. Model 1 is adjusted for the outcome measurement at baseline. Model 2 is adjusted for the outcome measurement at baseline and group affiliation (Paleolithic diet group or Paleolithic diet and exercise group). Model 3 is adjusted for the outcome measurement at baseline and change in weight. Abbreviations: BMI, body mass index. SBP, systolic blood pressure. DBP, diastolic blood pressure. HDL, High-density lipoprotein. LDL, Low-density lipoprotein.

## Data Availability

Not applicable.

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
