# Peer review of "Using a Paleo Ratio to Assess Adherence to Paleolithic Dietary Recommendations in a Randomized Controlled Trial of Individuals with Type 2 Diabetes"

_nutrients, 2021, doi:10.3390/nu13030969_

Round 1

Reviewer 1 Report

This is a secondary analysis of an RCT that has been published a couple of years ago. Although not stated explicitly, the manuscript suggests that the original study was an RCT testing the effect of a Palaeo diet in patients with type 2 diabetes. This is a bit misleading because the randomisation was for the exercise component, not for the Palaeo diet and thus there was no diet control group. Maybe this can be indicated more clearly in this manuscript. However, for the analysis described in this manuscript, this methodological issue is less important, because the data are analysed here as in an observational study. A disadvantage for this type of analysis is the small number of participants (N=27), but the authors acknowledge this and take it into account in their analyses.

The authors analysed whether the adherence to the Palaeo diet at week 12 was associated with the anthropometric and metabolic outcome variables at week 12 with (model2) or without (model 1) adjustment for the baseline value of the outcome variable. I am wondering whether the baseline value of the Palaeo ratio also plays a role. Maybe the authors could check this and add this to the manuscript.

In the discussion the authors suggest that the effect on the triglyceride levels of the Palaeo diet could be due to the higher fish consumption. Although this may be true, it is also well known that a higher carbohydrate intake is associated with higher triglyceride levels and this may also contribute to the effect.

The conclusion only focuses on the effects that are independent of weight loss. This seems to me a bit limited, why not also mention the relationships between the Palaeo index and the anthropometric variables and SBP?

Please check the reference list. Author names are missing.

Minor comments:

In Table 1 it is indicated that 19 participants used metformin and 3 did not use anti-diabetes medication. This does not add up to the total number of 27.

use superscript in R2 (R2)and kg/m2 (kg/m2).

The reduction in systolic blood pressure seems very large. Is this correct?

Author Response

Dear editor and reviewers of the manuscript

Thank you for giving us the possibility to resubmit our manuscript nutrients-1072149. We are most grateful for the helpfull comments from the reviewers. We have now revised our paper according to their suggestions. Below you find the reviewers comments in italics with our responses in regular style. We have specified the lines where the text has been changed and used the “Track Changes” function in the manuscript.

Dear authors,

nutrition is a big problem for patients with chronic disease, in particular for those with diabetes. Your conclusions are interesting but some points should be taken into consideration:

  1. Line 37: are you sure that scientific evidence is lacking?

We agree that evidence is not really lacking but rather inconclusive as many diets have evidence for improved cardiovascular risk factors and glucose control. We have changed the sentence in line 37-40.

  1. Line 57-61: paleolithic diet versus what? Please clarify which kind of studies were considered in the meta-analysis; which control group were considered?

In the meta-analysis, the Paleolithic diets was compared to several different control diets. We have now added examples of the control diets to the manuscript (line 60-62).

  1. you report on the Paleolithic ratio; I would be interested to know the amount of the caloric intake at the beginning and at the end of the study

We have added a new figure to the manuscript (figure 2) that shows the calorie intake for each participant at the beginning and the end of the study. We have added the calorie intake at baseline and at 12 weeks as median and IQR to the results section (line 186-188).

  1. can you rule out that the improvement in the outcomes is related to the clinical trial effect rather than to the increase in the Paleolithic index?
  2. Basically, I think that your conclusions are biased by the lack of a control group. Furthermore, you should take into consideration that the improvement can be due to reduced intake of calories or to the clinical trial effect.  

We have now added the results of an observational group to the paper (line 147-156, 230-237). The participants of that group were measured at baseline and after 12 weeks. However, this group was not randomized. Moreover, it is not possible of statistical reasons to include the observational group together with the intervention groups into the same regression analysis when investigating the results of the Paleo ratio at 12 weeks. The reason is that the Paleo ratio at 12 weeks is not normally distributed because all participants in the observational group have low values and all participants in the intervention groups have high values. We tested many transformations of this variable but none makes the variable sufficiently normally distributed.

The results of the observational group show that those individuals did not change much in neither Paleo ratio, body weight, blood pressure or plasma-triglycerides between the baseline measurements and the 12-weeks measurements. We therefore conclude that the effect we see in this paper is not a clinical trial effect. However, we totally agree that a part of the effect we see in this paper is caused by the reduced calorie intake and the weight loss. Certainly, for systolic blood pressure, weight loss and reduced calorie intake seems to be more important than compliance to the Paleolithic diet. According to our statistical analysis in regards to triglycerides, the Paleolithic diet seems to play an important part. However, that does not rule out that also the reduced calorie intake and the weight loss is very important. We have changed the conclusion of our paper to take your valuable comment into consideration.

This is a secondary analysis of an RCT that has been published a couple of years ago. Although not stated explicitly, the manuscript suggests that the original study was an RCT testing the effect of a Palaeo diet in patients with type 2 diabetes. This is a bit misleading because the randomisation was for the exercise component, not for the Palaeo diet and thus there was no diet control group. Maybe this can be indicated more clearly in this manuscript. However, for the analysis described in this manuscript, this methodological issue is less important, because the data are analysed here as in an observational study.

We agree with you that the study reported in this paper is an observational study. We have now stated that more clearly in the beginning of the methods section (line 80-81).

A disadvantage for this type of analysis is the small number of participants (N=27), but the authors acknowledge this and take it into account in their analyses.

The authors analysed whether the adherence to the Palaeo diet at week 12 was associated with the anthropometric and metabolic outcome variables at week 12 with (model2) or without (model 1) adjustment for the baseline value of the outcome variable. I am wondering whether the baseline value of the Palaeo ratio also plays a role. Maybe the authors could check this and add this to the manuscript.

We added a new model (called Model 1b in the methods section) which adjusted for Paleo ratio at baseline (line 165-167). Using this new model we found that adjusting for Paleo ratio at baseline did not change the effect of the Paleo ratio at 12 weeks on the variables SBP or triglycerides (line 165-167, 226-230).

In the discussion the authors suggest that the effect on the triglyceride levels of the Palaeo diet could be due to the higher fish consumption. Although this may be true, it is also well known that a higher carbohydrate intake is associated with higher triglyceride levels and this may also contribute to the effect.

We absolutely agree that the lower carbohydrate intake may be the cause for the decrease of the plasma triglyceride levels in our study. We have added this (together with a reference) to the discussion section (line 266-267).

The conclusion only focuses on the effects that are independent of weight loss. This seems to me a bit limited, why not also mention the relationships between the Palaeo index and the anthropometric variables and SBP?

We have rewritten the conclusion and now start with the results on the anthropometric variables and SBP (line 328-331).

Please check the reference list. Author names are missing.

We have added the missing author names to the reference list.

Minor comments:

In Table 1 it is indicated that 19 participants used metformin and 3 did not use anti-diabetes medication. This does not add up to the total number of 27.

use superscript in R2 (R2)and kg/m2 (kg/m2).

We have corrected the mistakes.

The reduction in systolic blood pressure seems very large. Is this correct?

The results reported in our manuscript are correct. The large reduction might be caused by the small number of participants (n=27) and by the fact that half of the study population was taking part in an exercise intervention.

Thank you for considering our manuscript for publication!

We are looking forward to hearing from you!

Yours sincerely,

Alexander Mårtensson and Julia Otten

Reviewer 2 Report

Dear authors,

nutrition is a big problem for patients with chronic disease, in particular for those with diabetes. Your conclusions are interesting but some points should be taken into consideration:

  1. Line 37: are you sure that scientific evidence is lacking?
  2. Line 57-61: paleolithic diet versus what? Please clarify which kind of studies were considered in the meta-analysis; which control group were considered?
  3. you report on the Paleolithic ratio; I would be interested to know the amount of the caloric intake at the beginning and at the end of the study
  4. can you rule out that the improvement in the outcomes is related to the clinical trial effect rather than to the increase in the Paleolithic index?
  5. Basically, I think that your conclusions are biased by the lack of a control group. Furthermore, you should take into consideration that the improvement can be due to reduced intake of calories or to the clinical trial effect.  

Author Response

(The authors gave the same response as above.)

Reviewer 3 Report

The paper analyzes whether the degree of adherence to the Paleolithic diet was associated with the changes in BMI, fat mass, blood pressure, triglycerides, HbA1c, and fasting glucose observed in a randomized controlled trial using Paleolithic diet and exercise in individuals with type 2 diabetes. In order to assess the proportion of Paleolithic food items in the diet of the individuals at baseline and after 12 weeks, the authors used a Paleo ratio calculated by dividing the energy content (kcal) of the Paleolithic food items by the total energy intake (kcal).

To test the association between the Paleo ratio and the effect of the dietary intervention, the authors used multiple linear regression via different adjusted models.

This study is a secondary analysis of a previous randomized controlled trial in which subjects with type 2 diabetes eating a Paleolithic diet for 12 weeks decreased their weight, fat mass, blood pressure, triglycerides, HbA1c, and fasting glucose. In the current article, the authors conclude that a higher degree of adherence to the Paleolithic diet is associated with improved triglyceride levels independent of weight loss among individuals with type 2 diabetes.

Although the impact of this article may be important in clarifying the controversial and sometimes contradictory results of the Paleolithic diet on some parameters by controlling the degree of adherence to the diet, the authors should improve certain aspects of the discussion and clarify some methodology and result sections for a better understanding and assessment. A review of the following aspects would be advisable:

- Authors should explain in a deeper way the objective and purpose of the observational group both in the study design section and in its specific subsection.

- The redaction style and number of references should be reviewed (for example, lines 59-61, and reference (21) in line 696).

- The study population includes participants of a wide range of ages (30-70 in males, and undefined postmenopausal age-70 in females). The authors do not refer to the possible physiological changes associated with aging that may affect some of the parameters analyzed, regardless of the dietary intervention.

- In the study design, participants were randomized to two groups according to whether their exercise regimens were supervised or not. However, it is not clear if all of them performed some kind of exercise (lines 77-78) or only one group (table 1). In addition, in the later sections, authors do not refer clearly to the objective of this distinction for the development and conclusion of the study.

- It is advisable to explain in more detail the different models used for the regression analysis, both in the 2.8 section and in the results and discussion (lines 216-225), especially regarding to the aim and significance of Model 1b (lines 161-162).

- Although intervention studies with Paleolithic diet rarely include recommendations for specific macronutrients proportion rather than diet composition, it is obvious from the authors’ results that there is a clear change towards a hypocaloric diet restriction (lines 181-183). It is not clear if this restriction affects more to one group than to the other. Nevertheless, it would be advisable a deeper analysis and discussion to remark that the observed effects are due to the composition of the diet and not only to a reduction in the caloric intake.

- Food items included in the dietary intervention for the different groups are sometimes confusing, since some of them are not clearly defined (e.g. vegetable products/vegetable dishes) and other ones are not excluded to facilitate adherence to the dietary intervention. This and other limitations might explain some effects and disfigure the results.

Author Response

Dear editor and reviewers of the manuscript

Thank you for giving us the possibility to resubmit our manuscript nutrients-1072149. We have addressed the reviewer’s valuable comments below. The reviewer’s comments are in regular style and our responses in italics.

- Authors should explain in a deeper way the objective and purpose of the observational group both in the study design section and in its specific subsection.

The observational group was added to show that the results of the intervention group differ from those of the observational group. This has now been stated more clearly in the materials and methods section bellow subheading 2.1 (line 86-87) and bellow subheading 2.7 (line 162-164) and in the discussion line 308-310.

- The redaction style and number of references should be reviewed (for example, lines 59-61, and reference (21) in line 696).

Line 58-59, 62. There was an error in our references that has now been adjusted.

- The study population includes participants of a wide range of ages (30-70 in males, and undefined postmenopausal age-70 in females). The authors do not refer to the possible physiological changes associated with aging that may affect some of the parameters analysed, regardless of the dietary intervention.

We agree that the inclusion criteria for age were wide. However, the youngest included participant was 44 years old and the oldest was 69 years old. We have taken your point in consideration – but we believe that the age range of 25 years has affected our regression analyses only to a small extent.

- In the study design, participants were randomized to two groups according to whether their exercise regimens were supervised or not. However, it is not clear if all of them performed some kind of exercise (lines 77-78) or only one group (table 1). In addition, in the later sections, authors do not refer clearly to the objective of this distinction for the development and conclusion of the study.

We now describe the two groups more clearly in the method section (Line 82-84) and the abstract (line 19). Group affiliation was used as a variable in our regression analyses. However, this variable had never a significant impact on the outcome variable. We have now added this information to the results section, line 253-255.

- It is advisable to explain in more detail the different models used for the regression analysis, both in the 2.8 section and in the results and discussion (lines 216-225), especially regarding to the aim and significance of Model 1b (lines 161-162).

We explained the models more clearly in the methods section, line 178-188. We have clarified where model 1b is presented in the results (line 279).

- Although intervention studies with Paleolithic diet rarely include recommendations for specific macronutrients proportion rather than diet composition, it is obvious from the authors’ results that there is a clear change towards a hypocaloric diet restriction (lines 181-183). It is not clear if this restriction affects more to one group than to the other. Nevertheless, it would be advisable a deeper analysis and discussion to remark that the observed effects are due to the composition of the diet and not only to a reduction in the caloric intake.

The median weight loss was 7.1 kg in both groups without difference between groups. We have added this information to the beginning of the results section, line 207-208. This suggests that the decrease in caloric intake was similar in both groups, at least on a group level over the time period of 12 weeks. To disentangle if diet quality measured with the Paleo ratio or caloric restriction/weight loss cause the improvement in blood pressure and triglycerides we use regression model 3 where both the Paleo ratio and weight loss are part of the model (Table 3), we showed this in the result section (line 253-255).

- Food items included in the dietary intervention for the different groups are sometimes confusing, since some of them are not clearly defined (e.g. vegetable products/vegetable dishes) and other ones are not excluded to facilitate adherence to the dietary intervention. This and other limitations might explain some effects and disfigure the results.

“Vegetable dishes” are vegetable products that contain non-Paleolithic food groups, e.g. dairy. “Vegetable products” are vegetable products only including Paleolithic food groups. We have added this information to table 2.

Thank you for your help to improve our manuscript!

Best regards,

Alexander Mårtensson and Julia Otten

Reviewer 4 Report

The authors present the revised version of their manuscript, which reports the results of a secondary observational post-hoc analysis of their RCT on the effects of supervised or unsupervised exercise on top of a "paleolithic diet".

The overall paper is well written, concise and easy to read.

The already included corrections and additional information have improved the paper.

Nevertheless, there are some points which need to be addressed:

1) Line 69: The citation should be (22), not (21).

2) The definition of a "paleolithic diet" in a contemporary interventional context is almost always flawed. Europeans of the stone age did not have access to potatoes, beef, pork, chicken, cocoa, coffee or sauces. The authors have tried to adjust their selection of paleolithic foods by implementing limits for certain foods (such as nuts, coffee, eggs etc.). However, either the introduction or the discussion should clearly state, that before the achievements of agriculture and animal domestication, meat intake was restricted to coast-line seafood, worms, bugs, larvae etc., and venison of all kinds (birds, boar, rodents, deer...). Foods of non-European origin should be clearly described as a modern-time adaptation of the genuine paleolithic diet, which are required for practical reasons and in order to achieve sufficient compliance. Maybe use the term "modernised paleolithic diet" to cover your nutritional adaptations to the Cro-Magnon original.

3) The implementation of a "paleo ratio" appears as a very ingenious idea about how to summarize compliance to a dietary pattern, which is predominantly defined by a selection of very diverse, specific foods rather than nutrients or their ratio. The results of the analysis are plausible, the metabolic improvements are at least at some level attributed to the paleo ratio by the means of causality. Weight loss explains a lot, but not everything.

I wonder, if any other single variable, describing the quality of the opposing diets (paleo vs. non-paleo), would facilitate a similar prediction of metabolic improvements. Does a protein/carbohydrate ratio or a MUFA/SFA ratio generate weaker results? By showing a superiority of the "paleo ratio" over these (or other suitable) ratios you could strengthen the argument, that the concept of the "modernised paleolithic diet" at all is a valid approach to improve diet quality with only secondary consideration of nutrient composition. Clearly, you should use a competitive ratio which actually reflects the previously reported dietary changes of both intervention groups, which surprisingly - as seen in the original paper from 2017 - seem to be rather different. (smaller baseline energy intake, loss in fiber and almost no protein increase in the exercise group etc.)

4) Table 3: Decimals should be shortened in accordance to the precision of the raw data.

5) Abstract and introduction should clarify, that the subjects were "asked" to follow a ("modernised") paleolithic diet, rather than stating that they "ate" it (either before or after initiation of the study). The current version implies full compliance, which is not true.

Author Response

Dear editor and reviewers of the manuscript

Thank you for giving us the possibility to resubmit our manuscript nutrients-1072149. We have addressed the reviewer’s valuable comments below. The reviewer’s comments are in regular style and our responses in italics.

1) Line 69: The citation should be (22), not (21).

Line 74, this is now corrected.

2) The definition of a "paleolithic diet" in a contemporary interventional context is almost always flawed. Europeans of the stone age did not have access to potatoes, beef, pork, chicken, cocoa, coffee or sauces. The authors have tried to adjust their selection of paleolithic foods by implementing limits for certain foods (such as nuts, coffee, eggs etc.). However, either the introduction or the discussion should clearly state, that before the achievements of agriculture and animal domestication, meat intake was restricted to coast-line seafood, worms, bugs, larvae etc., and venison of all kinds (birds, boar, rodents, deer...). Foods of non-European origin should be clearly described as a modern-time adaptation of the genuine paleolithic diet, which are required for practical reasons and in order to achieve sufficient compliance. Maybe use the term "modernised paleolithic diet" to cover your nutritional adaptations to the Cro-Magnon original.

We agree that the foods eaten during the Paleolithic age highly depend on the geographical location. We have added information regarding this to the introduction (line 47) and to the discussion (line 391-396). There is no 100% “real” Paleolithic diet as it depends on a number of factors. We agree that the Paleolithic diet eaten by our participants would not have been possible during the Paleolithic era.

3) The implementation of a "paleo ratio" appears as a very ingenious idea about how to summarize compliance to a dietary pattern, which is predominantly defined by a selection of very diverse, specific foods rather than nutrients or their ratio. The results of the analysis are plausible, the metabolic improvements are at least at some level attributed to the paleo ratio by the means of causality. Weight loss explains a lot, but not everything.

I wonder, if any other single variable, describing the quality of the opposing diets (paleo vs. non-paleo), would facilitate a similar prediction of metabolic improvements. Does a protein/carbohydrate ratio or a MUFA/SFA ratio generate weaker results? By showing a superiority of the "paleo ratio" over these (or other suitable) ratios you could strengthen the argument, that the concept of the "modernised paleolithic diet" at all is a valid approach to improve diet quality with only secondary consideration of nutrient composition. Clearly, you should use a competitive ratio which actually reflects the previously reported dietary changes of both intervention groups, which surprisingly - as seen in the original paper from 2017 - seem to be rather different. (smaller baseline energy intake, loss in fiber and almost no protein increase in the exercise group etc.)

We have analysed the association between our outcome measurements and the MUFA/SFA-ratio, the PUFA/SFA-ratio and the protein/carbohydrate ratio respectively. Triglyceride levels were highly associated with the MUFA/SFA and PUFA/SFA ratios, independently of weight loss.
Added information in the Method section (line 140-146, 189-196), the result section (line 262-277, figure 3,4, table 4), the discussion (line 327-344), to our conclusion (line 410) and to the abstract (line 28-31).

4) Table 3: Decimals should be shortened in accordance to the precision of the raw data.

We have now changed the number of decimals accordingly.

5) Abstract and introduction should clarify, that the subjects were "asked" to follow a ("modernised") paleolithic diet, rather than stating that they "ate" it (either before or after initiation of the study). The current version implies full compliance, which is not true.

Line 18, 71. This is now clearly stated.

Thank you for your help to improve our manuscript!

Best regards,

Alexander Mårtensson and Julia Otten

Round 2

Reviewer 2 Report

Dear authors,

thank you for replies. Did you try to include the caloric intake reduction into the regression model?
